# Enhanced Stability of Dimethyl Ether Carbonylation through Pyrazole Tartrate on Tartaric Acid-Complexed Cobalt–Iron-Modified Hydrogen-Type Mordenite

**DOI:** 10.3390/molecules29071510

**Published:** 2024-03-28

**Authors:** Guangtao Fu, Xinfa Dong

**Affiliations:** Guangdong Provincial Key Laboratory of Green Chemical Product Technology, School of Chemistry and Chemical Engineering, South China University of Technology, Guangzhou 510640, China; ce202120124494@mail.scut.edu.cn

**Keywords:** hydrogen-type mordenite, DME carbonylation, pyrazole tartrate, cobalt–iron, Bronsted acid sites

## Abstract

In this study, pyrazole tartrate (Pya·DL) and tartaric acid (DL) complexed with cobalt–iron bimetallic modified hydrogen-type mordenite (HMOR) were prepared using the ion exchange method. The results demonstrate that the stability of the dimethyl ether (DME) carbonylation reaction to methyl acetate (MA) was significantly improved after the introduction of Pya·DL to HMOR. The Co∙Fe∙DL-Pya·DL-HMOR (0.8) sample exhibited sustainable stability within 400 h DME carbonylation, exhibiting a DME conversion rate of about 70% and MA selectivity of above 99%. Through modification with the DL-complexed cobalt–iron bimetal, the dispersion of cobalt–iron was greatly enhanced, leading to the formation of new metal Lewis acidic sites (LAS) and thus a significant improvement in catalysis activity. Pya·DL effectively eliminated non-framework aluminum in HMOR, enlarged its pore size, and created channels for carbon deposition diffusion, thereby preventing carbon accumulation and pore blockage. Additionally, Pya·DL shielded the Bronsted acid sites (BAS) in the 12 MR channel, effectively suppressing the side reactions of carbon deposition and reducing the formation of hard carbon deposits. These improvements collectively contribute to the enhanced stability of the DME carbonylation reaction.

## 1. Introduction

Ethanol, highly regarded for its cleanliness and environmental friendliness, is a crucial energy source that can substitute or augment traditional fuels, thereby contributing to the mitigation of fossil fuel overconsumption [1]. The rapid growth of the fuel ethanol market has promoted the development of production methodologies embracing alternative carbonaceous sources, such as coal, biomass, or natural gas syngas. Notably, the process involved in the carbonylation of DME to produce MA, followed by hydrogenation to yield ethanol, has garnered significant attention from researchers because of its exceptional atom economy, high selectivity, and potential for widespread industrial implementation [2,3,4]. DME coupling with CO over acidic zeolites facilitates non-noble metal catalysis and a halide-free process to upgrade the widely available C1 intermediates into high value-added MA, which can be facilely hydrogenated to ethanol [5,6,7,8,9]. Thus, DME carbonylation is also considered the key step in ethanol synthesis, and it has attracted considerable attention over the past decade.

Iglesia et al. [10] found that HMOR zeolite exhibits unparalleled carbonylation activity in DME carbonylation reactions due to its unique skeletal composition. HMOR comprises a parallel 12-membered ring (12 MR) (0.70 × 0.67 nm) and 8 MR (0.57 × 0.26 nm) channels interconnected through 8 MR side pockets (0.48 × 0.34 nm) [11,12]. Notably, the carbonylation of DME mainly occurs within the 8 MR channel of HMOR, benefitting from its unique quantum domain-limiting effect. In contrast, the 12 MR channels housing Bronsted acid (BAS) serve as carbon-accumulating reactive sites prone to carbon deposition, thereby hastening the deactivation of HMOR during carbonylation [13]. To improve carbonylation activity and stability, it is necessary to manipulate the BAS distribution in HMOR precisely or to target the shielding of BAS in the 12 MR channel.

Scientists have been working to regulate the structure of HMOR in order to enhance its efficacy and stability in DME carbonylation. Li et al. [14] hydrothermally synthesized cerium-containing MOR samples. The substitution of aluminum in the zeolite framework with Ce^3+^ species increased the concentration of BAS in the 8 MR, enhancing carbonylation activity. Wang et al. [15] used the ion exchange method to modify HMOR with transition metals such as Ni, Co, Cu, Zn, and Ag, resulting in enhanced DME conversion and MA selectivity in the majority of the samples. Liu et al. [16] utilized trimethylamine cation (TMA^+^) for HMOR ion exchange, maintaining catalytic activity during three rounds of experiments. However, the specific surface area decreased significantly with each exchange, with it reaching less than 1/10 of the original surface area after three exchanges, accompanied by reduced DME conversion rates. In another study, Liu et al. [17] employed plate-like alkyl imidazolium ions for selective ion exchange in HMOR. The introduction of 1,3-dimethyl imidazolium ions allowed for the selective removal of BAS in the 12 MR channel, substantially improving the stability and activity of HMOR.

At present, although the stability of HMOR has been enhanced to some extent by shielding the BAS in the 12 MR channel, the regeneration of BAS still occurs in 12 MR, which poses a challenge in maintaining the prolonged high carbonylation activity and stability of HMOR. Ma et al. [18] found that the introduction of Co^2+^ into the 8 MR pore channel could improve the adsorption of CO and DME molecules, thereby enhancing reactivity. Zhou et al. [19] showed that incorporating Fe into the initial hydrothermal gel could reduce the intensity and density of BAS in the 12 MR, mitigating coke formation. Alternatively, pyrazole is a smaller nitrogen-containing heterocyclic compound with a molecular diameter of 0.434 nm. It would be more difficult for it to pass through the 8 MR tunnel and easier for it to enter the 12 MR tunnel to shield the active sites of carbon deposits [20]. Moreover, weakly basic pyrazole reacts with hydrochloric acid to form an aqueous solution of pyrazole hydrochloride (Pya∙HCl), which can achieve the same function of removing non-framework aluminum in HMOR as nitric acid [21]. Liu et al. [20] found that Pya∙HCl could selectively enter 12 MR in the ion exchange process, and, in turn, the stability of the modified HMOR was improved. Tartaric acid is highly acidic, and its boiling point is as high as 399.3 °C, which is much higher than the reaction temperature of DME carbonylation. Compared with other pyrazole salts, the use of pyrazole tartrate to modify HMOR could significantly retard pyrazole desorption during the reaction, making BAS difficult to regenerate in the pore channel and effectively suppressing the side reaction of carbon deposition on 12 MR, thereby substantially improving the stability of HMOR. In this study, combined with the characteristics and synergistic effect of tartaric acid-complexed cobalt–iron and pyrazole tartrate modification, a DME carbonylation catalyst with enhanced activity and better stability was successfully prepared through the incorporation of pyrazole tartrate and tartaric acid-complexed cobalt–iron bimetal modification on HMOR using the ion exchange method. The structure and properties of HMOR before and after modification were investigated using XRD, BET, NH_3_-TPD, Py-IR, FT-IR, and TG, and the fundamental reasons for the improvement in the activity and stability of modified HMOR catalysts are preliminarily explained.

## 2. Results and Discussion

### 2.1. Catalytic Performance

The unmodified hydrogen-type mordenite (HMOR) was prepared from ammonia-type mordenite (NH_4_MOR) via calcination. The samples Co∙Fe∙DL-HMOR and Co∙Fe∙DL-Pya·DL-HMOR (*x*) (*x* = 0.4, 0.6, 0.8, 1.0) were synthesized using the ion exchange method (Section 3.1).

Figure 1a depicts the DME conversion curves over time in the carbonylation reaction of HMOR, Co∙Fe∙DL-HMOR, and Co∙Fe∙DL-Pya∙DL-HMOR (*x*) (*x* = 0.4, 0.6, 0.8, 1.0) catalysts. As illustrated in Figure 1a, the DME conversion rate of unmodified HMOR reaches 35.5%. In contrast, the DME conversion rate significantly increases to 95% for Co∙Fe∙DL-HMOR, which is two-fold higher than the unmodified HMOR. However, these two catalysts were rapidly deactivated in the DME carbonylation reaction after 6 h. With further modification of the catalysts with Pya∙DL via ion exchange, even though the DME conversion rate of Co∙Fe∙DL-Pya∙DL-HMOR (*x*) slightly decreased compared to Co∙Fe∙DL-HMOR, the stability of DME carbonylation was significantly improved. As the concentration of Pya·DL gradually increased, the DME conversion rate exhibited a trend of first increasing and then decreasing. Interestingly, the DME conversion rate of the samples Co∙Fe∙DL-Pya·DL-HMOR (*x*) (*x* = 0.4, 0.6, 1.0) showed a slight downward trend with a prolonged reaction time. Notably, the DME conversion rate of Co∙Fe∙DL-Pya·DL-HMOR (0.8) consistently displayed an upward trend and exceeded that of Co∙Fe∙DL-Pya·DL-HMOR (0.6) at 180 h, indicating higher activity and better stability.

Figure 1b illustrates the curves of MA selectivity over time in the carbonylation reaction of HMOR, Co∙Fe∙DL-HMOR, and Co∙Fe∙DL-Pya∙DL-HMOR (*x*) (*x* = 0.4, 0.6, 0.8, 1.0) catalysts. Although the MA selectivity of Co∙Fe∙DL-HMOR was enhanced compared to the unmodified HMOR, the MA selectivities of HMOR and Co∙Fe∙DL-HMOR declined rapidly after reaching their peak. Co∙Fe∙DL-Pya∙DL-HMOR (*x*) displayed significantly enhanced MA selectivity after ion-exchange modification with Pya∙DL, with it maintaining a stable level of more than 99%.

In order to further investigate the performance of Co∙Fe∙DL-Pya∙DL-HMOR (0.8), a 400 h experiment was conducted. Figure 2 depicts the DME conversion and MA selectivity for the carbonylation reaction over the Co∙Fe∙DL-Pya∙DL-HMOR (0.8) catalyst over the 400 h experimental period. The Co∙Fe∙DL-Pya∙DL-HMOR (0.8) catalyst exhibited exceptional stability over the 400 h testing period. The DME conversion rate steadily increased, reaching about 73%, and it did not show a declining trend during the experimental period. Simultaneously, the MA selectivity remained consistently above 99%, confirming the sustained high activity and stability of the catalyst. Moreover, a comparison of the DME carbonylation performance of the HMOR modified via selective shielding and the removal of BAS within the 12 MR in recent years is provided in Appendix A. Compared with the other modified HMOR samples, Co∙Fe∙DL-Pya∙DL-HMOR (0.8) not only has higher DME carbonylation activity but also exhibits superior stability and a longer service life. 

### 2.2. XRD Analysis

Figure 3 presents the XRD spectra of HMOR, Co∙Fe∙DL-HMOR, and Co∙Fe∙DL-Pya∙DL-HMOR (*x*) (*x* = 0.4, 0.6, 0.8, 1.0). All of the modified samples exhibited a typical HMOR structure, and no crystalline phase related to cobalt and iron was observed, indicating superior dispersion of cobalt and iron ions in the zeolite structure. However, the change in the peak intensity of the modified HMOR indicates the significant influence of the DL complex cobalt–iron bimetallic modification and the Pya·DL ion exchange on the crystal structure and crystallinity. The characteristic peak intensity of Co∙Fe∙DL-HMOR was significantly enhanced, suggesting an improvement in crystallinity, likely attributed to the complexation effect of DL. This prevents the incorporation of Co^2+^ and Fe^3+^ ions into the zeolite framework, thereby preventing the destruction of the zeolite structure, thus further enhancing the dispersion and orderliness of cobalt and iron [22]. However, after further ion exchange modification with Pya·DL, Co∙Fe∙DL-Pya·DL-HMOR (*x*) (*x* = 0.4, 0.6, 0.8) exhibits a slightly reduced intensity of characteristic peaks compared to Co∙Fe∙DL-HMOR. The intensity of the characteristic peaks is slightly diminished, possibly attributed to the introduction of Pya·DL ions, causing a certain degree of zeolite framework collapse and pore channel blockage [17]. Notably, the intensity of characteristic peaks in Co∙Fe∙DL-Pya∙DL-HMOR (1.0) was significantly weakened, signifying a substantial reduction in crystallinity, which may be attributed to the high concentration of Pya∙DL, causing pore clogging and skeletal collapse [17]. This result is consistent with the observed decrease in DME conversion and carbonylation activity during the carbonylation activity test.

In addition, XRD tests were conducted on spent Co∙Fe∙DL-Pya·DL-HMOR (0.8) samples, as presented in Appendix A. The spent Co∙Fe∙DL-Pya·DL-HMOR (0.8) sample shows similar characteristic diffraction peaks to the fresh sample, and no other related crystal phases of cobalt–iron are observed. The crystallinity of the spent sample is significantly reduced. This may be due to the formation of carbon deposits during the reaction, which causes blockage of the pore channels and a collapse of the skeleton [17].

### 2.3. BET Test

The specific surface area and pore size distribution of the unmodified HMOR, Co∙Fe∙DL-HMOR, and Co∙Fe∙DL-Pya∙DL-HMOR (*x*) (*x* = 0.4, 0.6, 0.8, 1.0) samples were determined through N_2_ adsorption–desorption experiments. As shown in Appendix A, all samples exhibited distinctive type I isothermal adsorption curves with evident hysteresis loops, indicating typical microporous and irregular mesoporous structures [23]. It is worth noting that the pore size distributions of the various samples differed greatly, as can be seen in Figure 4. The pores of approximately 0.67 nm in size are attributed to the primary 12 MR channel, while pores of around 0.51 nm in size correspond to the 8 MR channel [24,25]. The pore structure information of both the 8 MR and the 12 MR can be obtained for unmodified HMOR and Co∙Fe∙DL-HMOR, which indicates effective cobalt–iron ion dispersion on the Co∙Fe∙DL-HMOR [18,19]. However, it was observed that the pore size of the Co∙Fe∙DL-Pya∙DL-HMOR (*x*) (*x* = 0.4, 0.6, 0.8, 1.0) samples significantly increased with the increase in the concentration of Pya·DL. Furthermore, as the concentration of Pya·DL increased, the pore size gradually expanded. This can be attributed to the removal of non-skeletal aluminum from HMOR during acidic Pya∙DL treatment, leading to an expansion in pore size and the creation of additional mesoporous structures [26]. The increase in pore size facilitates the timely diffusion of deposited carbon from the zeolite pores, preventing further coalescence and pore blockage [27]. Consequently, the Co∙Fe∙DL-Pya∙DL-HMOR (*x*) (*x* = 0.4, 0.6, 0.8, 1.0) samples exhibit enhanced stability, aligning with the results of their carbonylation activity tests. Notably, the Co∙Fe∙DL-Pya∙DL-HMOR (1.0) sample showed a lack of discernible micropore distribution, potentially attributed to the introduction of excessive Pya∙DL ions, causing pore clogging and skeletal collapse, consistent with the carbonylation activity and XRD patterns.

Table 1 summarizes the BET-specific surface area and pore volume of the unmodified HMOR, Co∙Fe∙DL-HMOR, and Co∙Fe∙DL-Pya∙DL-HMOR (*x*) (*x* = 0.4, 0.6, 0.8, 1.0) samples. The Co∙Fe∙DL-Pya∙DL-HMOR (*x*) (*x* = 0.4, 0.6, 0.8, 1.0) samples exhibited a significant decrease in specific surface area and microporous volume. The observed changes are attributed to the substitution of H^+^ ions in the 12 MR pores with larger pyrazole ions [16]. Higher Pya∙DL concentrations result in increased BAS replacement in the 12 MR pores, which effectively suppress carbon accumulation reactions on the 12 MR and enhance catalyst stability. However, excessive Pya∙DL introduction leads to decreased surface area and pore volume, hindering reactant and product diffusion and causing a significant decline in catalytic activity.

### 2.4. NH_3_-TPD Analysis

Figure 5 illustrates the NH_3_-TPD plots of the unmodified HMOR, Co∙Fe∙DL-HMOR, and Co∙Fe∙DL-Pya·DL-HMOR (*x*) (*x* = 0.4, 0.6, 0.8) samples. Each sample exhibits two desorption peaks, a low-temperature peak (150–300 °C) associated with weak acid desorption and a high-temperature peak (400–700 °C) related to strong acid desorption [28]. Compared to the unmodified HMOR, the low-temperature desorption peaks for the Co∙Fe∙DL-HMOR and Co∙Fe∙DL-Pya·DL-HMOR (*x*) (*x* = 0.4, 0.6, 0.8) samples shifted to higher temperatures, indicating enhanced weak acidity resulting from DL-complexed cobalt–iron bimetallic and Pya·DL ion-exchange modification. This enhanced weak acidity further promotes carbonylation reactivity, which is consistent with the results of the carbonylation activity tests.

As can be seen in Figure 5, the high-temperature desorption peak for the Co∙Fe∙DL-HMOR sample slightly shifts to a lower temperature, suggesting that additional metal Lewis acidic sites (LAS) are generated after the modification treatment with the DL complex cobalt–iron bimetal, which enhances the acidic sites of the zeolite [29]. This increase in acidic sites contributes to the heightened conversion rate of DME, consistent with the results of the carbonylation activity test. Moreover, the high-temperature desorption peak center of the Co∙Fe∙DL-Pya·DL-HMOR (*x*) (*x* = 0.4, 0.6, 0.8) samples further shifts to lower temperatures with increasing Pya·DL concentration, which may be ascribed to the ion-exchange treatment with acidic Pya·DL aiding in the removal of non-skeletal aluminum from HMOR, resulting in a decrease in the strength of the strong acid [16]. The decrease in strong acid strength has the potential to inhibit side reactions, minimize carbon formation, and prolong the catalyst lifespan, consistent with the results of the carbonylation activity test.

### 2.5. Py-IR Analysis

Basic NH_3_ molecules can combine with BAS or LAS on the outer surface of mordenite and in the pores. However, NH_3_-TPD cannot determine whether the acidic sites come from the 8 MR or 12 MR of HMOR [30]. Pyridine, with a kinetic diameter of 0.585 nm, can enter the 12 MR channel of mercerized zeolite. However, it cannot enter its 8 MR channel [29,31]. Pyridine can only adsorb to the outer surface of HMOR and the acidic sites within the 12 MR pores [32]. Consequently, Py-IR analysis was utilized to provide a detailed examination and characterization of the acid content within the 12 MR pores. Figure 6 displays the Py-IR spectra of the unmodified HMOR and Co∙Fe∙DL-Pya·DL-HMOR (*x*) (*x* = 0.4, 0.6, 0.8) samples. Distinct peaks at 1540 cm^−1^ represent BAS, whereas peaks at 1450 cm^−1^ indicate Lewis acid sites (LAS), and the peaks at 1490 cm^−1^ arise from the combined influence of BAS and LAS [19,33]. The acid amounts in the 12 MR calculated from the peak areas at 1540 cm^−1^ and 1450 cm^−1^ reveal a significant reduction in the amount of BAS acid in the 12 MR of the Co∙Fe∙DL-Pya·DL-HMOR (*x*) samples (Table 2). Combined with the NH_3_-TPD data, one can conclude that the increased number of acid sites in the modified HMOR is derived from the LAS of the 8 MR; therefore, the carbonylation activity is enhanced. As the concentration of Pya·DL increases, the amount of BAS acid in the 12 MR progressively decreases. The Py-IR spectra indicate that the non-skeletal aluminum in the 12 MR of HMOR was removed via the ion exchange treatment with Pya·DL. Thus, the BAS, which was the active center of the carbon accumulation side reaction, was replaced by Pya·DL molecules. Higher concentrations of Pya·DL lead to improved shielding of BAS in the 12 MR, resulting in a substantial improvement in carbonylation stability, which is consistent with the NH_3_-TPD test results, carbonylation activity test, and stability test.

### 2.6. FT-IR Analysis

In order to further analyze the catalytic active species of Co∙Fe∙DL-Pya·DL-HMOR (0.8) during carbonylation and the changes in acid sites after the reaction, FT-IR tests were carried out on the fresh and spent Co∙Fe∙DL-Pya·DL-HMOR (0.8) samples, as shown in Figure 7. As can be seen in Figure 7a, the peaks at 3050 cm^−1^ and 2970 cm^−1^ correspond to the asymmetric stretching vibration of -CH_3_, which indicates that DME adsorbs on hydroxy aluminum (AlOH) to form methoxy [34]. The peak at 1680 cm^−1^ corresponds to the stretching vibration of acetyl zeolite intermediates, indicating that CO reacts with methoxy groups to form acetyl groups [2]. The peak at 1745 cm^−1^ corresponds to the stretching vibration of the reaction product MA, which indicates the productivity of MA after the reaction of acetyl groups with DME [31]. The peak at 1450 cm^−1^ corresponds to the stretching vibration of coke species, indicating that the reaction process is accompanied by carbon deposition side reactions [35]. The above results clearly indicate that methoxy and acetyl groups are the catalytic active species in the carbonylation reaction over the Co∙Fe∙DL-Pya·DL-HMOR (0.8) sample. 

As can be seen from Figure 7b, the peak at wavenumber 3660~3670 cm^−1^ corresponds to the -OH stretching vibration of BAS in mordenite, and the peak at 3720 cm^−1^ corresponds to the -OH stretching vibration of the terminal silanol group. Moreover, the characteristic bands of 3660~3670 cm^−1^ can be divided into high-frequency bands corresponding to the OH group in the 12 MR and low-frequency bands corresponding to the OH group in the 8 MR [36,37,38]. The intensity of the characteristic peak at 3720 cm^−1^ of the spent Co∙Fe∙DL-Pya·DL-HMOR (0.8) sample was significantly weakened, while the characteristic peak intensity at 3660~3670 cm^−1^ was evidently enhanced and moved toward the low-frequency direction. This indicates that the BAS levels of the 8 MR increased in the spent Co∙Fe∙DL-Pya·DL-HMOR (0.8) sample, meaning that the carbonylation activity of the catalyst will gradually increase with the progression of the reaction, which is consistent with the results of the stability test of the Co∙Fe∙DL-Pya·DL-HMOR (0.8) sample.

### 2.7. TG Analysis

It is generally believed that the BAS acid site of 12 MR is responsible for the formation of coke. DME reacts with methoxy adsorption states in 12 MR channels to form trimethoxyonium cations (TMO+), and TMO+ is generally considered to be the precursor of hydrocarbons [39]; therefore, BAS in the 12 MR channels will lead to rapid coke formation. Moreover, DME may only enter the active sites in the 8 MR hole through the 12 MR; thus, the 12 MR channel is very important for DME diffusion into and MA diffusion out of the active sites in the 8 MR hole [40,41]. In order to probe the carbon deposition in the 12 MR of the modified HMOR, we carried out TG tests on the fresh/spent HMOR and Co∙Fe∙DL-Pya·DL-HMOR (0.8), as shown in Figure 8.

Figure 8a illustrates the TG and DTG curves for the HMOR and Co∙Fe∙DL-Pya·DL-HMOR (0.8) samples. The weight loss observed below 250 °C corresponds to the evaporation of adsorbed water [16]. HMOR exhibits no significant weight loss peaks beyond 250 °C, indicating the removal of most internal organic templates during calcination. However, the DTG curves of the fresh Co∙Fe∙DL-Pya·DL-HMOR (0.8) samples do not exhibit a clear weight loss plateau at temperatures of 180–350 °C, suggesting the tight binding of pyrazole to HMOR, which is challenging to desorb after Pya·DL ion exchange modification. Moreover, a notable weight loss peak at 515 °C in the DTG curve of the unreacted Co∙Fe∙DL-Pya·DL-HMOR (0.8) is attributed to the combustion and carbonization of pyrazole [16].

Figure 8b illustrates the TG and DTG curves of HMOR after 10 h of reaction and the Co∙Fe∙DL-Pya·DL-HMOR (0.8) sample after 400 h of reaction. The weight loss rate of spent Co∙Fe∙DL-Pya·DL-HMOR (0.8) is lower than that of the spent HMOR. When the temperature exceeds 250 °C, the DTG curves reveal two weight loss peaks for the spent HMOR: a small peak at 335 °C from soft carbon combustion and a more prominent peak at 565 °C from hard carbon combustion [16]. This indicates that unmodified HMOR accumulates substantial hard carbon, which severely hinders the molecular diffusion process and leads to rapid catalyst deactivation. In contrast, the DTG curve of the spent Co∙Fe∙DL-Pya·DL-HMOR (0.8) shows only two small weight loss peaks above 250 °C, a smaller peak at 430 °C caused by pyrazole combustion and a larger peak at 580 °C caused by hard carbon combustion, and the combustion weight loss peak of soft carbon is almost invisible [16]. The less-soft and hard carbon-generated spent Co∙Fe∙DL-Pya·DL-HMOR (0.8) is attributed to the shielding of BAS in the 12 MR channel by pyridine following Pya·DL ion exchange modification. In summary, the introduction of Pya·DL into the 12 MR channel of HMOR effectively shields BAS, inhibiting carbon accumulation and enhancing carbonylation reaction stability.

## 3. Materials and Methods

### 3.1. Materials

A specific type of ammonia-treated mercerized zeolite with a silica–aluminum ratio of 15, designated as NH_4_MOR, was obtained from Wuhan Zhizhen Molecular Sieve Company (Wuhan, China). All chemicals and reagents, including Co(NO_3_)_2_·6H_2_O, Fe(NO_3_)_3_·9H_2_O, C_4_H_6_O_6_, and C_3_H_4_N_2_, are commercially available and were used directly as received.

### 3.2. Preparation of the Catalyst

The NH_4_MOR was heated in a muffle furnace at a rate of 3 °C min^−1^ and calcined at a temperature of 500 °C for 3 h in air. The product was hydrogen-treated mercerized zeolite, denoted as HMOR.

Firstly, 1.164 g Co(NO_3_)_2_·6H_2_O and 1.616 g Fe(NO_3_)_3_·9H_2_O were weighed and dissolved in 40 mL deionized water and stirred until the metal salt was completely dissolved. The obtained solution was a cobalt–iron bimetallic salt aqueous solution with a concentration of 0.2 mol/L and a molar ratio of cobalt to iron of 1:1. Afterward, under stirring, 0.008 moles of tartaric acid (DL) were added into 40 mL at a 0.2 mol∙L^−1^ concentration of cobalt–iron bimetallic salt aqueous solution (cobalt–iron molar ratio = 1:1) to obtain DL-complexed cobalt–iron bimetallic salt aqueous solution. Next, 2 g of NH_4_MOR was added to the bimetallic salt aqueous solution, which was stirred evenly. The obtained solution was then subjected to ion exchange at 80 °C in a water bath. After filtration, the resulting material was washed with deionized water 3–5 times, dried at 120 °C overnight, and calcined in air at 500 °C for 3 h. The final product was DL-complexed cobalt–iron modified HMOR, denoted as Co∙Fe∙DL-HMOR.

Pyrazole was added to DL to prepare a pyrazole tartrate solution (Pya∙DL) with a 1:1 molar ratio of Pya to DL. Subsequently, 2 g of Co∙Fe∙DL-HMOR was immersed in 40 mL of the Pya∙DL solution with concentrations of 0.4, 0.6, 0.8, and 1.0 mol∙L^−1^. The mixtures were thoroughly stirred, and ion exchange was conducted at 80 °C. After filtration, washing, and drying overnight at 120 °C, Co∙Fe∙DL-Pya∙DL-HMOR (*x*) was obtained. The value *x* represents the molar concentration of Pya∙DL (*x* = 0.4, 0.6, 0.8, 1.0) during the ion exchange process.

### 3.3. Characterization

The XRD patterns of the samples were recorded using a Bruker D8 Advance X-ray diffractometer with Cu Kα radiation (λ = 0.154 nm), operating at a current of 40 mA and a voltage of 40 kV. The scanning range was set between 5 and 50°. Before the test, 50 mg powder samples were smeared uniformly onto a sample holder to ensure a flat upper surface.

Nitrogen adsorption–desorption was conducted on the ASAP 2020 fully automatic physical adsorption instrument manufactured by the Micromeritics Company in the United States. The nitrogen adsorption–desorption process was executed at −196 °C. Before testing, the samples underwent degassing and pretreatment at 220 °C for 6 h. The specific surface area was determined using the BET method, while the micropore volume was calculated using the t-plot method.

The ammonia temperature-programmed desorption (NH_3_-TPD) test was carried out using the AutoChem II 2920 instrument manufactured by the Micromeritics Company in the Norcross, GA, USA, and TCD was used as the detector. A sample weighing 100 mg was placed in the heating zone of a quartz tube, purged with 30 mL·min^−1^ of helium, and heated to 210 °C. After constant-temperature water removal for 1 h, the temperature was lowered below 100 °C. Subsequently, NH_3_ was introduced at a flow rate of 30 mL·min^−1^ and maintained for 30 min to saturate the sample with NH_3_ adsorption. Following this process, helium was purged at 373 °C for 1 h. When the baseline signal corresponding to the mass spectrum stabilizes, the temperature is ramped from room temperature to 700 °C at a heating rate of 10 °C·min^−1^. The signal of NH_3_ desorbed on the catalyst during this process is detected, and, finally, the NH_3_ temperature-programmed desorption curve is obtained.

In the pyridine adsorption infrared (Py-IR) characterization test, pyridine was employed as the probe molecule, and the Vector 33-IR model Fourier-transform infrared spectrometer from Bruker, Ettlingen, Germany, was used to assess the acid amount and acid type of the sample. In general, a sample of about 15 mg was weighed, finely ground with an agate mortar, and compressed into cohesive flakes. The compacted sheet was inserted into an in situ cell capable of both heating and evacuation and equipped with CaF_2_ windows. Spectral analysis was performed on the sample using an MCT detector within the wavenumber range of 4000–1000 cm^−1^. Before pyridine adsorption, the sample was degassed at 400 °C for 1 h to purify the catalyst surface. Subsequently, a background scan was conducted when the temperature decreased to 150 °C. Pyridine was introduced into the inlet by employing an injector under N_2_ carrier gas for 0.5 h and was evacuated for 0.5 h, and then the sample was scanned, and its spectrum was recorded.

The FT-IR experiments were carried out on a Vertex 33-IR infrared spectrometer from Bruker, Germany. The samples were self-supported, and the samples were mixed with KBr at a mass ratio of 1:10 and fully ground three times, and, then, the test samples were pressed at 15 MPa to obtain homogeneous semi-transparent films, which were dried at 120 °C for 20 min. In total, 32 scans were used for the tests, with a resolution of 4 cm^−1^ and a scanning range of 4000~400 cm^−1^.

A German NETZSCH thermogravimetric analyzer (TG) was employed to assess carbon deposits and other organic matter in the samples. Approximately 8 mg of the sample was accurately weighed and placed in an Al_2_O_3_ ceramic crucible. Subsequently, under an airflow of 20 mL·min^−1^, the temperature was incrementally increased from 40 °C to 850 °C at a heating rate of 10 °C·min^−1^, and the mass change curve of the sample was recorded.

### 3.4. DME Carbonylation Reaction

The core reaction equation of the carbonylation reaction of DME is as follows:(1)$${\mathrm{CH}}_{3}{\mathrm{OCH}}_{3}+\mathrm{CO}\;\overset{cata/1.5\;MPa/215\;{}^{\circ}C}{\to }\;{\mathrm{CH}}_{3}{\mathrm{COOCH}}_{3}$$

The schematic diagram of the catalyst testing system is presented in Appendix A. The 0.25 g catalyst with a 40–60 mesh size was loaded into a stainless steel reactor with an inner diameter of 6 mm. Before the reaction, the catalyst was dried at 240 °C under a nitrogen flow of 20 mL·min^−1^. After the catalyst bed cooled to 215 °C, the reactants (4% DME, 76% CO, and 20% N_2_) were introduced into the reactor at a gas hourly space velocity of 4800 mL·g^−1^·h^−1^. The reaction pressure was then increased to 1.5 MPa. The effluents from the reactor were analyzed using an online Agilent 4890 gas chromatograph equipped with an FID detector and a capillary column (APPARATUS 0807242514). The conversion of DME (X_DME_) and the selectivity of each component (S_i_) were calculated based on the carbon balance principle and the corrected area normalization method.
(2)$$X_{\mathrm{DME}}=\frac{\sum_{k=2}^{5}A_{k}\cdot f_{M\left(k/\mathrm{DME}\right)}+\sum A_{C_{n}H_{m}}}{\sum_{k=1}^{5}A_{k}\cdot f_{M\left(k/\mathrm{DME}\right)}+\sum A_{C_{n}H_{m}}}\text{×}100\%$$
(3)$$S_{i}=\frac{A_{i}\cdot f_{M(i/\mathrm{DME})}}{\sum_{k=2}^{5}A_{k}\cdot f_{M\left(k/\mathrm{DME}\right)}+\sum A_{C_{n}H_{m}}}\text{×}100\%$$

In the formula, A_i_ is the chromatographic peak area of DME, methanol, ethanol, MA, and ethyl acetate components (i = 1, 2, 3, 4, 5), f_(M(i/DME))_ is the molar correction factor of component i relative to that of DME, and C_n_H_m_ is the hydrocarbon by-product.

## 4. Conclusions

In conclusion, a Pya·DL and DL complex cobalt–iron bimetallic modified HMOR catalyst with superior DME carbonylation activity and stability was successfully prepared using the ion exchange method in this study. Through the modification of the DL complexed cobalt–iron bimetal, the dispersion of cobalt–iron was greatly improved, and new metal Lewis acidic sites (LAS) were formed; thus, the activity of the catalyst significantly improved. Furthermore, the ion exchange of Pya·DL effectively removed non-skeletal aluminum in HMOR, resulting in an enlarged pore diameter and enhanced mesopore formation, which facilitated carbon deposition diffusion and prevented pore blockage. Simultaneously, the shielding of BAS by replacing equilibrium charge H^+^ in the 12 MR pore with larger pyrazole ions also inhibited carbon accumulation on the 12 MR and significantly enhanced the stability of the DME carbonylation reaction.

## Figures and Tables

**Figure 1 molecules-29-01510-f001:**
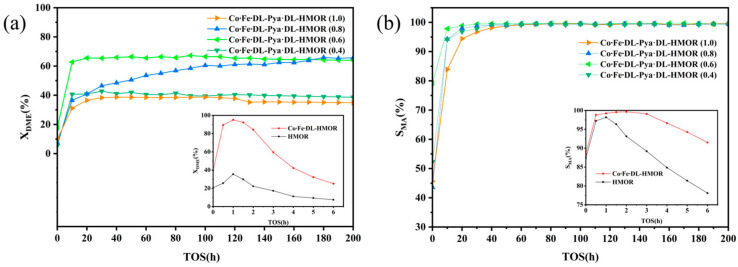
DME conversion rate (**a**) and MA selectivity rate (**b**) of HMOR, Co∙Fe∙DL-HMOR, and Co∙Fe∙DL-Pya·DL-HMOR (*x*) (*x* = 0.4, 0.6, 0.8, 1.0).

**Figure 2 molecules-29-01510-f002:**
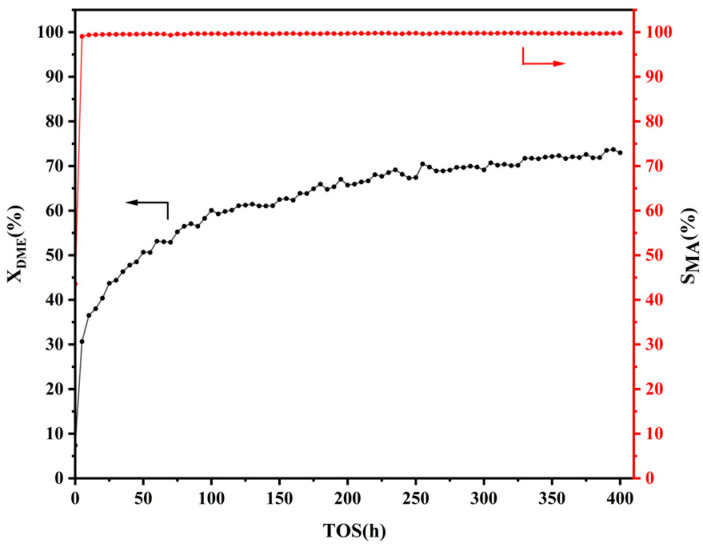
DME conversion and MA selectivity versus time for the Co∙Fe∙DL-Pya∙DL-HMOR (0.8) carbonylation reaction over 400 h.

**Figure 3 molecules-29-01510-f003:**
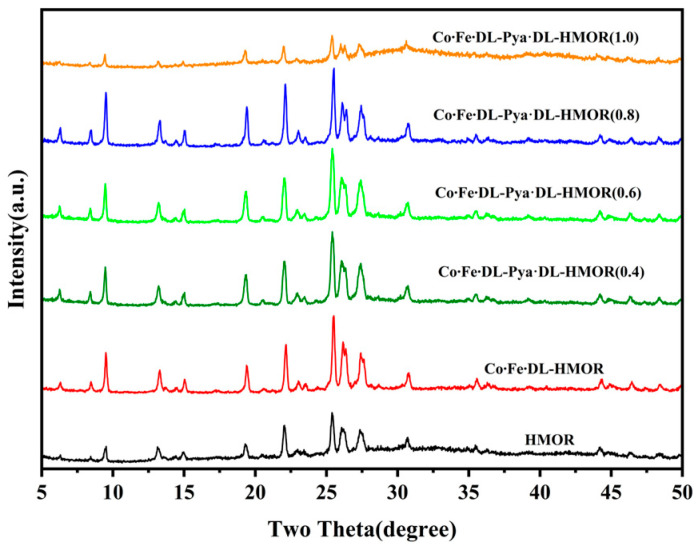
XRD spectrum of HMOR, Co∙Fe∙DL-HMOR, and Co∙Fe∙DL-Pya·DL-HMOR (*x*) (*x* = 0.4, 0.6, 0.8, 1.0).

**Figure 4 molecules-29-01510-f004:**
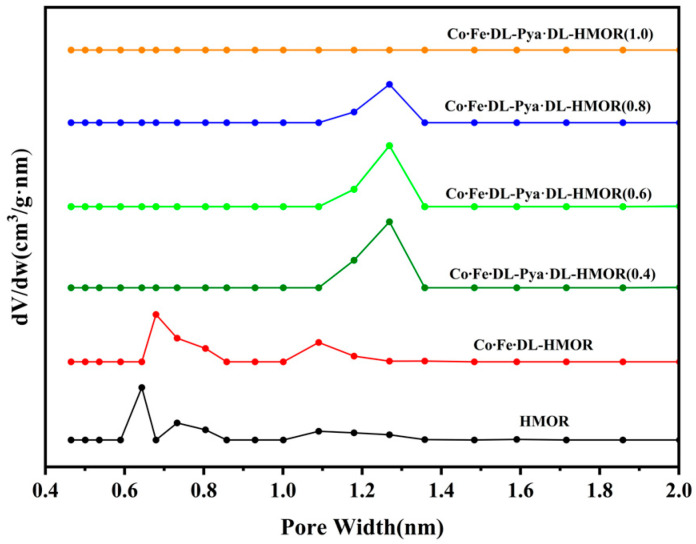
Pore size distribution of HMOR, Co∙Fe∙DL-HMOR, and Co∙Fe∙DL-Pya·DL-HMOR (*x*) (*x* = 0.4, 0.6, 0.8, 1.0).

**Figure 5 molecules-29-01510-f005:**
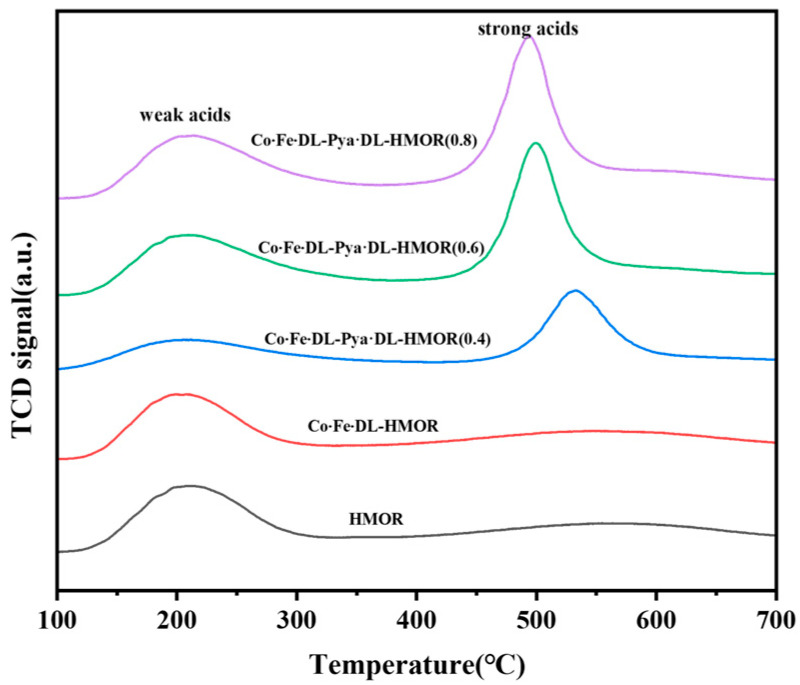
NH_3_-TPD profiles of HMOR, Co∙Fe∙DL-HMOR, and Co∙Fe∙DL-Pya·DL-HMOR (*x*) (*x* = 0.4, 0.6, 0.8).

**Figure 6 molecules-29-01510-f006:**
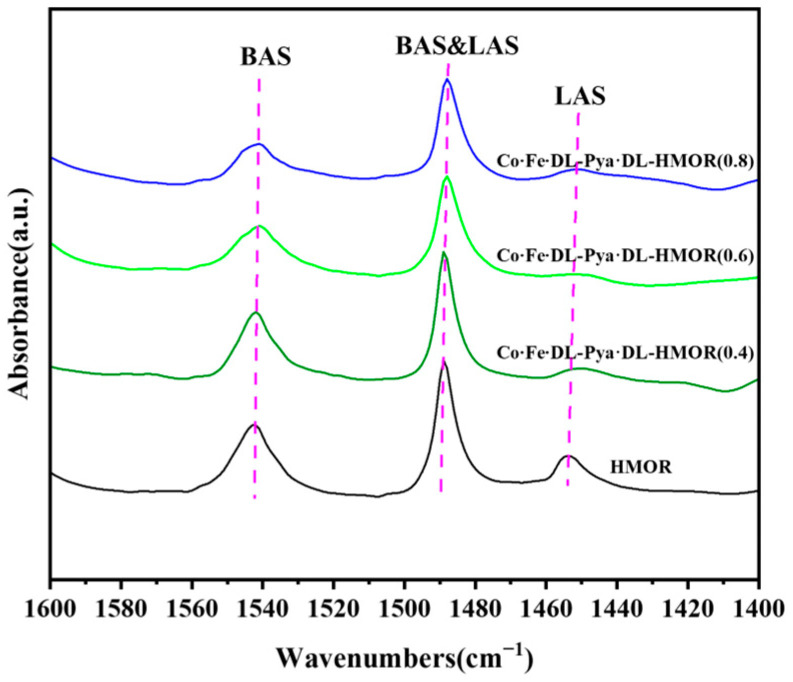
Py-IR spectra of HMOR and Co∙Fe∙DL-Pya·DL-HMOR (*x*) (*x* = 0.4, 0.6, 0.8).

**Figure 7 molecules-29-01510-f007:**
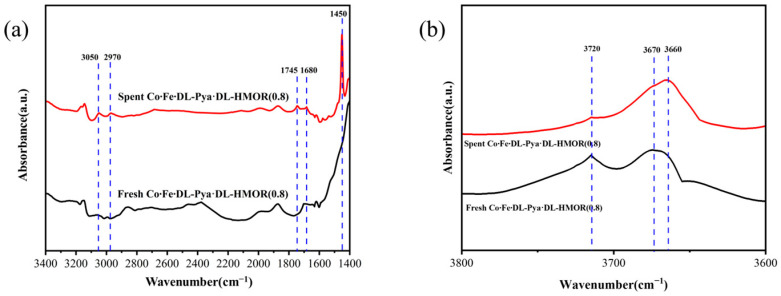
FT-IR spectra of the fresh and spent Co∙Fe∙DL-Pya·DL-HMOR (0.8) catalyst: (**a**) wavenumber in 1400~3400 cm^−1^ and (**b**) wavenumber in 3600~3800 cm^−1^.

**Figure 8 molecules-29-01510-f008:**
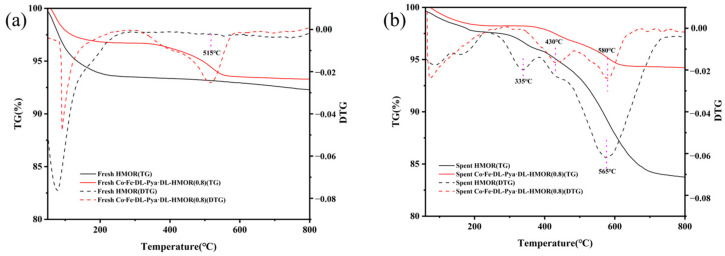
TG and DTG curves of (**a**) the fresh HMOR and Co∙Fe∙DL-Pya·DL-HMOR (0.8) samples and (**b**) the spent HMOR and Co∙Fe∙DL-Pya·DL-HMOR (0.8) catalysts.

**Table 1 molecules-29-01510-t001:** Specific surface area and micropore volume of HMOR, Co∙Fe∙DL-HMOR, and Co∙Fe∙DL-Pya·DL-HMOR (*x*) (*x* = 0.4, 0.6, 0.8, 1.0).

Sample	S_BET_/(m^2^/g)	V_total_/(cm^3^/g)
HMOR	352	0.257
Co∙Fe∙DL-HMOR	335	0.211
Co∙Fe∙DL-Pya·DL-HMOR (0.4)	238	0.171
Co∙Fe∙DL-Pya·DL-HMOR (0.6)	142	0.118
Co∙Fe∙DL-Pya·DL-HMOR (0.8)	53.1	0.087
Co∙Fe∙DL-Pya·DL-HMOR (1.0)	6.07	0.043

**Table 2 molecules-29-01510-t002:** Quantities of the acidic sites for HMOR and Co∙Fe∙DL-Pya·DL-HMOR (*x*) (*x* = 0.4, 0.6, 0.8).

Sample	BAS in the 12 MR/(µmol/g)	LAS in the 12 MR/(µmol/g)	BAS and LAS in the 12 MR/(µmol/g)
HMOR	237	57	294
Co∙Fe∙DL-Pya·DL-HMOR (0.4)	172	28	200
Co∙Fe∙DL-Pya·DL-HMOR (0.6)	113	18	131
Co∙Fe∙DL-Pya·DL-HMOR (0.8)	58	16	74

## Data Availability

The datasets generated and/or analyzed during the current study are available from the corresponding author upon reasonable request.

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
