# Peer review of "Enhanced Stability of Dimethyl Ether Carbonylation through Pyrazole Tartrate on Tartaric Acid-Complexed Cobalt–Iron-Modified Hydrogen-Type Mordenite"

_molecules, 2024, doi:10.3390/molecules29071510_

Round 1
Reviewer 1 Report
Comments and Suggestions for Authors
While it is difficult to indicate HMOR as an easily available raw material constituting a catalytic support, the information contained in the manuscript indicates both its interesting properties related to the porosity of this material, as well as its usefulness in the production of catalytic systems intended for the carbonylation reaction.
In the introduction, the authors try to convince us that the multi-stage synthesis of ethanol used as a fuel makes sense. I am not convinced by their attempts and unclear arguments. Especially the economic aspect of this ethanol synthesis path.
The introduction is written in a way that, in my opinion, does not encourage reading. The introduction should only outline the research goal against the background of achievements to date in this field, with an explanation of what improvements result from the research carried out by the authors. This part should be improved.
The presented results are interesting, supported by properly selected analytical methods. Many hours (200h / 400h) of tests of the obtained catalytic systems also speak in favor of the presented results
Although the DME to MA carbonylation reaction studied by the authors is simple, I suggest the authors show it in the manuscript.
Part "Results and Discussion” is written in an understandable manner. The authors explain the results of their research on improving the catalytic system for DME carbonylation, presenting the results of conversion and selectivity of the process - Figure 1.
I have no objections to analytical methods - XRD, BET, NH3-TPD, TG, Py-IR.
The description of the synthesis of the tested materials is also prepared properly.
A diagram of the catalyst testing system should be attached to the Supplementary Materials.
Reviewer 2 Report
Comments and Suggestions for Authors
The present manuscript by Fu and Dong describes a catalytic system for the carbonylation of dimethylether that appears to be better than previously reported ones. However, the work is quite incomplete in my view. I am an expert in carbonylation reaction and less expert min heterogeneous catalysis, so some pitfalls may escaper my attention, but there are many points that would need clarification.
1) The preparation of the catalyst is not described in a way that allow it reproduction. The authors mention a "cobalt-iron bimetallic salt aqueous solution" without even mentioning which salts they used and the oxidation state of the metal. Cobalt is present as cobalt(II) in most salts, but the oxidation state of iron (II or III) is not obvious at all. The counteranion will surely at least remain in the catalyst (even coordinated to the metal) and play an important role during the reaction.
2) All the characterization data refer to the as-prepared catalyst. However, the fact that the activity is initially very low and increases with time at least up to a certain point indicates that the catalytically active species is formed during the reaction. This is not surprising as CO should reduce the initially divalent (?) metals. The traditional catalytically active species for methanol carbonylation to acetic acid/methyl acetate is the reduced HCo(CO)4, even when CoCl2 or CoI2 are employed as a precatalyst, as done in the oldest technology. An IR spectrum of the catalyst after it has been employed may shed some light of the species that have been formed and an XRPD spectrum would identify if any metallic phase is formed. In general, a much more in depth characterization of the catalyst after the reaction should be performed.
3) The authors prepare a mixed Co/Fe, but never explain why they tried this combination. They also say that " In comparison with other pyrazole salts, the use of pyrazole tartrate complicates the desorption of pyrazole tartrate cations that enter the 12-MR and ion exchange with BAS………………….." However, no reference was given for this statement. If it a well known effect, a reference should be given. Otherwise, evidence should be given for this statement and for the choice to employ tartaric acid rather than any other acid.
In conclusion, the work is incomplete. Much data was reported to characterize a material which is not catalytically active, whereas no information is given on what it becomes when it has transformed into the catalytically active species. A manuscript in which a catalyst is prepared without any rationale and without any information of what it is when it acts adds no advancement to Science and should not be accepted as such.
Comments on the Quality of English LanguageThe quality of the English language is fine. "hydrothermal" at page 2 should be "hydrothermally"
Reviewer 3 Report
Comments and Suggestions for Authors
Review (major revision):
In the manuscript titled; “Enhanced Stability of DME Carbonylation through Pyrazole 2 Tartrate on Tartaric Acid-Complexed Cobalt-Iron Modified 3 HMOR” by the authors Guangtao Fu and Xinfa Dong, the authors studied a dual metal modified hydrogen-type mordenite, promoted with pyrazole tartrate and tartaric acid.
Introduction:
The introduction part is well written with some minor corrections in English necessary.
Results and Discussion:
The captions and writing in Fig 1 should be improved. The names shortened, the fonts increased,…
The authors should clarify why DL addition improves crystallinity of HMOR. There are logical inconsistencies in this paragraph. The writing/English should be improved to resolve this. (e.g. Line 140-141: …, which prevents disrupts the zeolite structure by the incorporation of Co2+ and Fe3+ ions into the zeolite framework).
In nitrogen sorption, the desorption curve does not return to the same volume as the adsorption isotherm. This indicates an error in measurement. Additionally, the diameter of the nitrogen molecule is approximately 0.3 nm. The “Median pore width/(nm)” column of Table 1 is therefore completely wrong.
Why is there no low temperature peak for the Co·Fe·DL-Pya·DL-HMOR(0.4) sample?
In ammonia-TPD analysis the authors state that additional acid sites are generated. They negate this in the next chapter, where they state that acid site shielding occurs. The authors should elaborate on this further.
Both TG curves should have the same y-axis, so that they are easier to compare. The authors should clarify how BAS is associated with carbon accumulation.
Materials and Methods:
What was the atmosphere for the calcination of ammonia-treated mercerized zeolite.
The ASAP 2020 is manufactured and sold by the Micromeritics Company.
Why was ammonia, in NH3-TPD, purged at 373 °C? What detector was used?
The term “air velocity” is not appropriate. The authors should define either gas velocity or gas hourly space velocity?
Conclusion:
The conclusion part is well written, but requires some editing.
Comments on the Quality of English LanguageThe English is quite bad in some parts of the manuscript. In some cases, it is hard to understand what the authors meant by a statement as there are logical inconsistencies.
Round 2
Reviewer 2 Report
Comments and Suggestions for Authors
The authors fully replied to my questions and concerns by adding more data and discussion. I now support publication of the manuscript in the present form.